



# Spectral Calibration of the MethaneAIR Instrument

Carly Staebell[1], Kang Sun[1,2], Jenna Samra[3], Jonathan Franklin[4], Christopher Chan Miller[3], Xiong Liu[3], Eamon Conway[3], Kelly Chance[3], Scott Milligan[5], and Steven Wofsy[4,6]

[1]Department of Civil, Structural and Environmental Engineering, University at Buffalo, Buffalo, NY, USA
[2]Research and Education in Energy, Environment and Water Institute, University at Buffalo, Buffalo, NY, USA
[3]Harvard-Smithsonian Center for Astrophysics, Cambridge, MA, USA
[4]School of Engineering and Applied Sciences, Harvard University, Cambridge, MA, USA
[5]Headwall Photonics, Bolton, MA, USA
[6]Department of Earth and Planetary Sciences, Harvard University, Cambridge, MA, USA

**Correspondence:** Kang Sun (kangsun@buffalo.edu)

**Abstract.** MethaneAIR is the airborne simulator of MethaneSAT, an area-mapping satellite currently under development with the goal of locating and quantifying large anthropogenic point $CH_4$ sources as well as diffuse basin-scale emissions. Built to closely replicate the forthcoming satellite, MethaneAIR consists of two imaging spectrometers. One detects $CH_4$ and $CO_2$ absorption around 1.65 and 1.61 $\mu$m, respectively, while the other constrains the optical path in the atmosphere by detecting $O_2$

absorption near 1.27 $\mu$m. The high spectral resolution and stringent retrieval accuracy requirements of greenhouse gas remote sensing in this spectral range necessitate a reliable spectral calibration. To this end, on-ground laboratory measurements were used to derive the spectral calibration of MethaneAIR, serving as a pathfinder for the future calibration of MethaneSAT. Stray light was characterized and corrected through Fast Fourier Transform (FFT)-based Van Cittert deconvolution. Wavelength registration was examined and found to be best described by a linear relationship for both bands with a precision of ∼0.02

spectral pixel. The instrument spectral spread function (ISSF), measured with fine wavelength steps of 0.005 nm near a series of central wavelengths across each band, was oversampled to construct the instrument spectral response function (ISRF) at each central wavelength and spatial pixel. The ISRFs were smoothed with a Savitzky-Golay filter for use in a lookup table in the retrieval algorithm. The MethaneAIR spectral calibration was evaluated through application to radiance spectra from an instrument flight over the Colorado Front Range.

## 1 Introduction

Methane ($CH_4$) is an influential greenhouse gas due to its high global warming potential, which is estimated to be 56–105 times higher than that of carbon dioxide for a 20-year time period (Howarth, 2014). Given that approximately 60% of global $CH_4$ emissions are anthropogenic (Saunois et al., 2020), the identification and subsequent reduction of these sources represent a

significant opportunity for climate change mitigation (Zhang et al., 2020). In quantifying atmospheric $CH_4$, space-based obser-



vation is a powerful tool due to its ability to provide regular coverage on a variety of spatial scales. Currently, the Greenhouse gases Observing SATellite (GOSAT) and TROPOspheric Monitoring Instrument (TROPOMI) collect $CH_4$ abundance data at a global scale (Yoshida et al., 2011; Hu et al., 2016). These missions have effectively captured regional emission trends as well as significant point sources, but their somewhat coarse spatial resolutions ($\sim$10 km for GOSAT and 7 km for TROPOMI)

limit source differentiation and location (Varon et al., 2020). Target-mode satellite instruments may offer much higher spatial resolution for the observation of individual high-emitting facilities within small areas. For example, the GHGSat-D instrument targets $CH_4$ sources at 50 m resolution within a 12 km$^2$ area (Varon et al., 2019, 2020; Jervis et al., 2020). Space-based observations at a scale larger than that of GHGSat-D but smaller than that of TROPOMI are not currently operational, but may contribute to a more robust greenhouse gas monitoring system.

MethaneSAT is a push broom imaging satellite under development that is designed to operate at such an intermediate scale. The mission by MethaneSAT, LLC, a subsidiary of the Environmental Defense Fund, is planned to launch in 2022 (MethaneSAT, LLC, 2020). MethaneSAT aims to characterize basin-scale, diffuse $CH_4$ emissions through a wide swath of $\sim$ 200 km$^2$ and at the same time locate and quantify large point sources within each target area that is typically $200 \times 140$ km$^2$ (Benmergui et al., 2020; MethaneSAT, LLC, 2020). MethaneSAT detects $CH_4$ absorption around 1.65 $\mu$m and $CO_2$ absorption

near 1.61 $\mu$m with one spectrometer. In order to constrain the optical path in the atmosphere, a second spectrometer is dedicated to detect the $O_2$ $a^1\Delta_g$ band around 1.27 $\mu$m. Although the $O_2$ $A$ band ($\sim$0.76 $\mu$m) has been commonly used for this purpose, the $O_2$ $a^1\Delta_g$ band may be more advantageous given its close proximity to the $CH_4$ band and recent advances in airglow correction (Sun et al., 2018; Bertaux et al., 2020).

MethaneAIR is the airborne simulation instrument for MethaneSAT. It has been designed to replicate the forthcoming satel-

lite as closely as possible. The MethaneAIR instrument was integrated in the National Science Foundation (NSF) Gulfstream V (GV) aircraft operated by the National Center for Atmospheric Research (NCAR). Two engineering flights were conducted in the Colorado Front Range in November 2019. Similarly, airborne simulators have been built and analyzed in support of other future satellites, such as GEOstationary Trace gas and Aerosol Sensor Optimization (GEO-Taso) (Nowlan et al., 2016), GEOstationary Coastal and Air Pollution Events (GEO-CAPE) Airborne Simulator (GCAS) (Nowlan et al., 2018), and Air-

borne Compact Atmospheric Mapper (ACAM) (Liu et al., 2015). These three instruments have been used as testbeds for the geostationary Tropospheric emissions: Monitoring of pollution (TEMPO) instrument. The test flights from airborne simulators aid in algorithm development while also providing valuable scientific data (Nowlan et al., 2016, 2018; Liu et al., 2015). Indeed, in-flight observations can supplement current ground-based and satellite remote sensing, as is the goal of the airborne SWIR spectrometers Methane Airborne MAPper (Gerilowski et al., 2011), and GreenHouse gas Observations of the Stratosphere and

Troposphere (GHOST) (Humpage et al., 2018).

The high spectral resolution and stringent retrieval accuracy requirements of greenhouse gas remote sensing in the short-wave infrared (SWIR) band necessitate a reliable spectral calibration. One of the most important tasks of spectral calibration is characterization of the instrument spectral response function (ISRF), the response of a spectral pixel to photons at different wavelengths. An accurate understanding of the ISRF is crucial for the retrieval of greenhouse gas abundance from the observed

radiance spectra. Analysis of column averaged dry-air mole fractions of carbon dioxide ($X_{CO_2}$) as measured by the Orbiting



Carbon Observatory-2 (OCO-2) revealed that uncertainties due to ISRF dominated the total error in land nadir observations (Connor et al., 2016). For $CH_4$ retrieval in particular, simulations of TROPOMI $X_{CH_4}$ retrievals have indicated high sensitivity to errors in the ISRF (van Hees et al., 2018). The ISRFs are usually characterized through preflight measurements and parameterized with functions ranging in complexity from a simple Gaussian (Munro et al., 2016; Hamidouche and Lichtenberg, 2018) or super-Gaussian (Beirle et al., 2017) to a tailored, weighted sum of multiple functions (van Hees et al., 2018; Liu et al., 2015). In the absence of a satisfactory functional form, a lookup table may be used to describe the ISRF (Day et al., 2011; Lee et al., 2017). Laboratory measurements for ISRF characterization are also used to determine wavelength registration of spectral pixels (Lee et al., 2017).

The ISRF is often determined only within a few spectral sampling intervals away from the central wavelength. However, the spectrum from a given spatial pixel can contain spatial stray light from the other spatial pixels and spectral stray light from all spatial pixels. The TROPOMI SWIR band stray light was measured extensively in an on-ground calibration campaign. A near-real time stray light correction was incorporated in the TROPOMI operational data processor by approximating the stray light by a pixel-independent far-field kernel and an additional kernel representing the main reflection. This process reduces stray light error, thus increasing gas-column retrieval accuracy (Tol et al., 2018).

This work presents the spectral calibration of the MethaneAIR instrument, which serves as the pathfinder of the calibration efforts of the MethaneSAT instrument, given the close similarity in spectral characteristics between MethaneAIR and MethaneSAT. The structure of this paper is organized as follows. The MethaneAIR instrument and its integration on the GV aircraft is briefly summarized in Section 2. The ISRF and stray light calibration setups are described in Section 3. The stray light correction procedure is described in Section 4. Section 5 presents the construction of the ISRF look-up table, and Section 6 presents the determination of wavelength registration based on the ISRF calibration data set. The spectral calibration is applied in the MethaneAIR spectral fitting algorithm in Section 7, followed by the conclusion in Section 8.

## 2 MethaneAIR instrument overview

MethaneAIR consists of two customized imaging spectrometers from Headwall Photonics (part number 1003A-20507 for the $O_2$ spectrometer and 1003A-20507 for the $CH_4$ spectrometer), each with a 1.3 megapixel InGaAs camera from Princeton Infrared Technologies. The airborne instrument provides similar spectroscopy to MethaneSAT with higher spatial resolution, although with a significantly smaller swath width. Instrument specifications are listed in Table 1.

Figure 1 shows a representation of the light path within each spectrometer. Light entering through the foreoptic is focused onto the entrance slit. The slit is then imaged onto the focal plane through an Offner spectrometer with a convex holographic grating, resulting in a 2D image with spatial information along one dimension and wavelength along the other. In Figure 1, the spectral dimension of the image is up/down, and the spatial dimension is into/out of the page. The optical design provides sub-pixel smile and keystone distortion and relatively uniform focus across wavelength and field angle. Anti-reflection coatings and high grating efficiency provide an optical transmittance of 37–39% in the $CH_4$ channel and 45–47% in the $O_2$ channel. The $CH_4$ channel has a polarization sensitivity of 5–10% and the $O_2$ channel has a polarization sensitivity of 20–25%.





**Table 1.** MethaneAIR specifications.

| Optical and Detector | | Spectral and Spatial | |
| --- | --- | --- | --- |
| Focal length (mm) | 25 | $O_2$ passband (nm) | 1236–1319 |
| F-number | 3.5 | $O_2$ dispersion (nm/pixel) | 0.08 |
| Entrance slit width ($\mu$m) | 34 | $O_2$ spectral FWHM (nm) | 0.22 |
| Optical transmittance (%) | $\geq 37$ | $CH_4$ passband (nm) | 1592–1680 |
| Polarization sensitivity (%) | $\leq 25$ | $CH_4$ dispersion (nm/pixel) | 0.1 |
| FPA dimensions (spectral $\times$ spatial pixels) | $1024 \times 1280$ | $CH_4$ spectral FWHM (nm) | 0.3 |
| Pixel pitch ($\mu$m) | 12 | Plate scale (°/pixel) | 0.0275 |
| Quantum efficiency below 1650 nm | $> 0.7$ | Field of view (°) | 23.7 |
| Frame rate (Hz) | 10 | Swath width (km) at 12 km altitude | 5.05 |
| Readout noise (e-), typical | 35 | Cross-track pixel (m) at 12 km altitude | 5.76 |
| Dark current (e-/s/pixel) | $< 8,500$ | Along-track pixel (m) | $\approx 25$ |

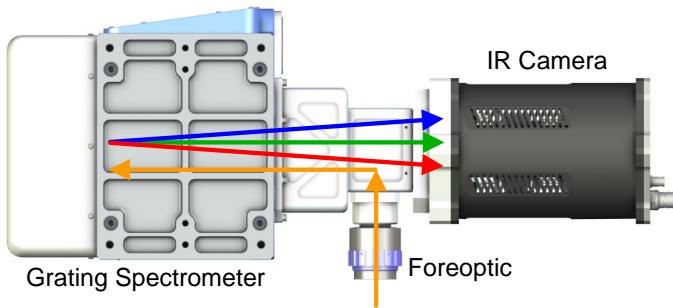

**Figure 1.** Light path inside each spectrometer.

The infrared camera used in each channel is the 1280SCICAM from Princeton Infrared Technologies. The InGaAs focal plane provides greater than 0.7 quantum efficiency (QE) below 1650 nm. The QE begins to roll off above this wavelength, reaching a minimum of about 0.15 at the 1680 nm end of the $CH_4$ passband. The focal plane operates at 0°C, which provides a reasonable compromise between dark current and the temperature-sensitive long-wavelength cutoff. The 1024 columns and 1280 rows of the focal plane array (FPA) correspond to spectral and spatial pixels as shown in Figure 5. Only spatial pixel indices 135–997 and 308–1170 out of 1–1280 are illuminated by the slit for the $CH_4$ and $O_2$ bands, respectively.

Initial MethaneAIR research flights were performed aboard the NSF GV aircraft. To simplify aircraft integration, the two MethaneAIR spectrometers are mounted side by side in a single instrument rack (Figure 2), which is isolated from aircraft vibration by wire isolators. Each spectrometer was internally aligned from foreoptic to focal plane by Headwall, and the two spectrometers were co-boresighted to within 1° when they were mounted in the rack. For $CH_4$ and $CO_2$ measurements, the spectrometers observe out of an 18 inch viewport on the bottom of the GV, using a 25 mm wide angle lens (23.7° field of view). Both panes of glass in the viewport window were anti-reflection coated by L&L Optical Services. The spectral reflectivity of



all four surfaces was measured from 400 to 1700 nm by the coating manufacturer. The resulting window transmittance is a smooth function ranging from 99.7% to 98.1% in the MethaneAIR spectral range. A 180 degree rotation of the instrument rack allows the $O_2$ spectrometer to observe out of the overhead viewport in order to image the airglow, using an 85 mm lens that provides a 7° field of view.

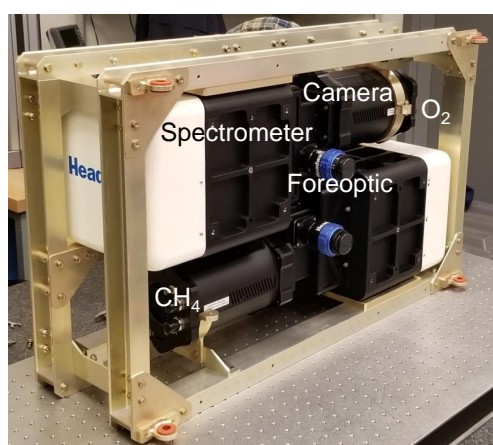
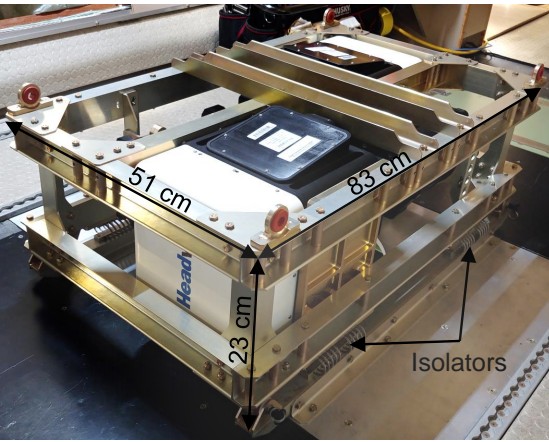

**Figure 2.** MethaneAIR instrument rack, side view (left) and down-looking configuration in the GV aircraft (right).

## 3 Calibration measurements

In an effort to reproduce the mechanical and thermal environment experienced during flight, MethaneAIR was mounted during laboratory calibration activities on a rack in its downward viewing orientation and was controlled to just above room temperature by the same thermal housing used aboard the GV (Figure 3a-b). Calibration equipment (including an integrating sphere and a collimator) were placed under the rack pointing upward. Each spectrometer collected measurements for stray light and ISRF calibration, as described in the next two subsections. In addition, flat fields were taken using the integrating sphere coupled with a broadband lamp behind a variable aperture.

The integrating sphere, model #OL 455-8SA-2 from Optronic Laboratories, has an overall diameter of 8 inches and a 2 inch diameter output port (Figure 3c). The spectral radiance at the output port was calibrated by the manufacturer every 10 nm between 350 nm and 2500 nm. During the MethaneAIR flat field measurements, the light level was tuned from zero to just beyond detector saturation in 40 steps by adjusting the aperture area. The aperture area was tied to the manufacturer calibration value using a photodetector mounted on the wall of the sphere. Flat field data were taken at exposure times of 50, 100, and 150 ms, matching the exposure times used in flight and in the ISRF calibration. The resulting radiometric calibration curves were fitted by fifth-order polynomials with the intercept forced to be zero. The resultant coefficients were used to correct pixel-to-pixel non-uniformity in the stray light and ISRF data. A separate linear (gain-only) calibration was used to flag defective pixels. Bad pixels were identified as those with a dark value more than 3-sigma from the mean or a gain value outside thresholds





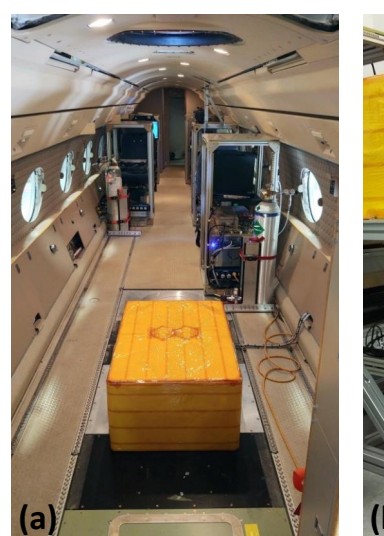 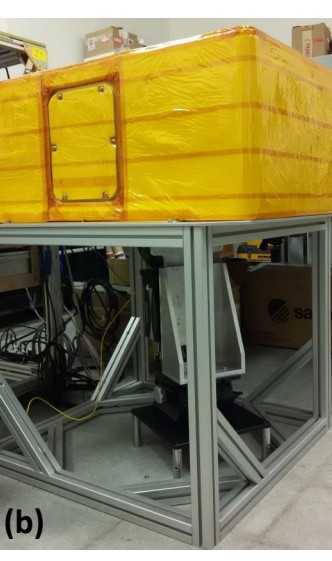 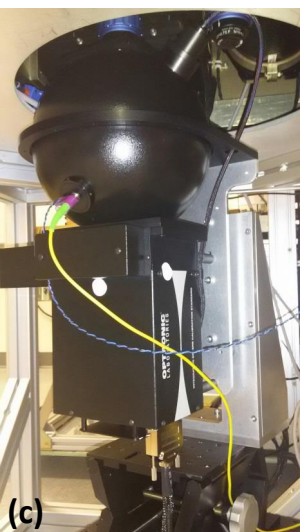 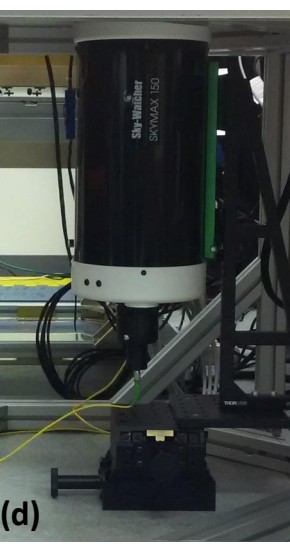

**Figure 3.** During flight (a) and calibration (b), the instrument is mounted in the same orientation and controlled to the same temperature by the yellow thermal housing. An integrating sphere (c) is used to perform non-uniformity and ISRF calibration, while a collimator (d) is used for stray light measurements.

determined by visual inspection of the gain distribution. Bad pixels made up 0.19% and 0.055% of the active area of the $O_2$ and $CH_4$ FPAs, respectively.

### 3.1 Stray light measurements

Stray light measurements were made by systematically illuminating individual points on the focal plane and quantifying the light detected elsewhere on the detector. Preliminary stray light measurements used a 150 mm diameter f/12 Maksutov-Cassegrain telescope (Figure 3d) to collimate the incoming light. A 100 $\mu$m pinhole placed at the focus of the telescope was illuminated using fiber-coupled tunable lasers (SANTEC TSL-550s; one for $O_2$ and one for $CH_4$). The laser line width is 40 MHz, three orders of magnitude lower than the instrument spectral resolution, and hence the laser is considered as a delta function in wavelength space. At the slit, the image of the pinhole fit within $12 \times 12$ $\mu$m (equivalent to one FPA pixel).

At each sampled spatial position the tunable laser was stepped across the passband in increments of 0.5 nm. The collimator was mounted on a goniometer stage and manually repositioned to sample three angles along the slit ($0°$, $-7°$, $+9°$ in $CH_4$ band; $0°$, $-5°$, $+10°$ in $O_2$ band). Exposure times of 10 ms, 100 ms, and 1000 ms were combined for high dynamic range, and one additional exposure was made at 1000 ms while increasing the laser power by a factor of 10 (Figure 4a-d). Background measurements were made by temporarily closing a shutter internal to the tunable laser and subtracted from each individual exposure.

Future improvements to the measurement setup will include automated tilt and translation stages to address many more field angles and an all-reflective collimator to avoid stray reflections from the refractive corrector plate. The pinhole will be replaced

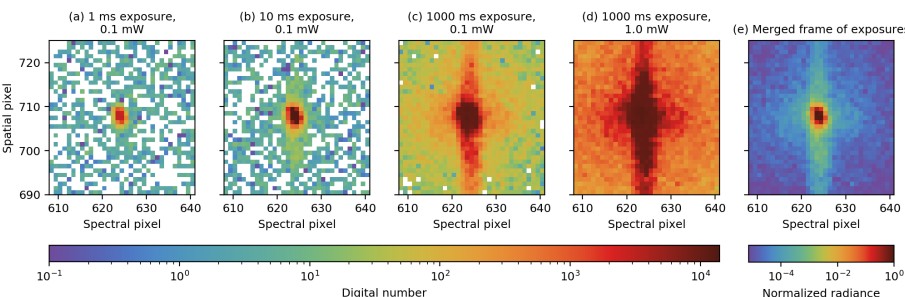

**Figure 4.** Individual frames with different exposure times in ms and listed laser power in mW (a-d) are combined to create the resulting normalized merged frame (e). At shorter exposure times, the peak is well defined, but the floor is dominated by measurement noise. Longer exposure times allow for characterization of the floor, but the peak and its surrounding area become saturated.

with a 100 $\mu$m slit oriented perpendicular to the spectrometer entrance slit, in order to fill the width of the spectrometer slit while providing a point source in the across-track dimension.

## 3.2 ISRF measurements

ISRF measurements used the same tunable lasers as the stray light measurements. Each laser was coupled to the integrating sphere (Figure 3c) in order to uniformly illuminate the slit at a single wavelength. An 8000 RPM vibration motor was attached to the fiber near the integrating sphere to avoid coherence effects in the image. The $O_2$ laser was stepped from 1247 nm to 1317 nm in increments of 7 nm, and the $CH_4$ laser was stepped from 1593 nm to 1679 nm in increments of at most 10 nm (see Figure 5). The vicinity of each center wavelength was finely sampled by scanning the laser $\pm 0.1$ nm from the central wavelengths in steps of 0.005 nm. The 1247 nm central wavelength step was discarded in following analysis because it is right at the edge of the laser wavelength cut off. In the $CH_4$ band, the laser power was increased at long wavelengths to compensate for the decrease in QE. Both spectrometers recorded data with a fixed exposure time of 50 ms.

## 4 Stray light correction

Stray light correction for MethaneAIR follows an approach similar to the method set forth by Tol et al. (2018) for the TROPOMI SWIR spectrometer. Preliminary processing of the stray light measurement data includes masking bad pixels and subtracting dark current. Radiometric calibration is applied to convert from digital number per second to radiance, and each frame is normalized by its corresponding laser power. Multiple frames at a given position on the FPA can then be combined into a single merged frame, as shown in Figure 4e. Merging different exposures allows for a more complete characterization of stray light structure since the peak is defined but the floor is incomplete at short exposures, and at longer exposures, the floor is defined while the peak area is saturated. A 2D Gaussian function is fitted to each merged frame to identify the central spatial and spectral position of the peak. The identified spectral peak positions were analyzed as a potential supplement to ISRF measurements for wavelength registration, but were ultimately found to be too noisy for this purpose. The partial illumination





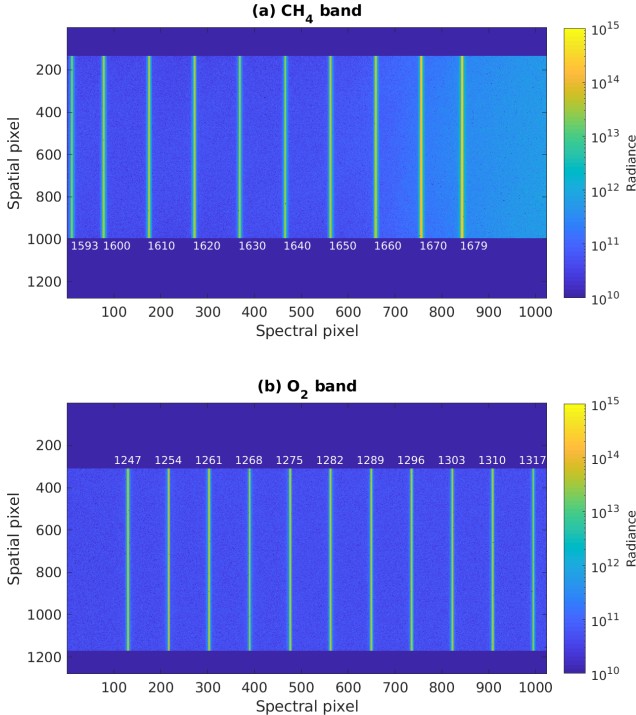

**Figure 5.** Responses of the focal plane arrays shown as calibrated radiance for the CH$_4$ (a) and O$_2$ band (b) over a range of laser wavelengths. The laser wavelengths in nm are labeled next to the corresponding slit images. The radiance is in photons s$^{-1}$ cm$^{-2}$ nm$^{-1}$ sr$^{-1}$.

of the slit resulting from the use of the pinhole in the measurement setup likely contributed to the noise by distorting the spectral response.

All merged frames are interpolated to a common grid of spatial and spectral pixels that are relative to peak position obtained from the 2D Gaussian fitting. The stray light structure observed in the merged frames is generally consistent for different positions on the FPA. The only notable exceptions are spatial stray light features that are up to $10^{-4}$ of the peak. These features, which appear in the tails of the spatial stray light profile, exhibit no apparent pattern relative to spatial position. That is, spatial stray light features at one spatial pixel were not observed in the profile measured at a nearby spatial pixel. Such inconsistency suggests that these features are not internal to the instrument, but likely originated from the reflections from the refractive corrector plate within the collimator. As such, data displaying what appear to be spatial artifacts of the test setup are removed via replacement with NaN values. Since stray light measurements were taken for three spatial positions, at least one other spatial position that does not exhibit the observed artifact still supplies data at the replaced point. After excluding these spurious spatial stray light data points, all merged frames are stacked together and the median is determined to produce a common kernel function for the entire FPA.

The median stray light kernels for both the CH$_4$ and O$_2$ bands are depicted in Figure 6, where it may be seen that the peaks are separated from the noise floor by over six orders of magnitude. For use in the stray light correction algorithm, the kernel





is normalized such that all elements sum to unity. A central area of 11 spatial pixels by 15 spectral pixels is then set equal
to zero. This window is determined by the extent of the ISRF in the spectral dimension and the width of the spatial response
function in the across-track dimension. The kernel is now referred to as the far-field kernel ($\mathbf{K}_{\mathrm{far}}$), which defines where stray
light correction will be applied.

The correction algorithm is rooted in the idea that a measured frame can be viewed as an ideal frame convolved with $\mathbf{K}_{\mathrm{far}}$.
Therefore, to correct the stray light, an iterative deconvolution algorithm is used, based on Van Cittert deconvolution (Tol et al.,
2018). The correction is a redistribution rather than a removal of light in a given frame. As given by Tol et al. (2018), the frame
($\mathbf{J}$) after iteration $i$ is

$$\mathbf{J}_i = \frac{\mathbf{J}_0 - \mathbf{K}_{\mathrm{far}} \otimes \mathbf{J}_{i-1}}{1 - \sum_{k,l}(\mathbf{K}_{\mathrm{far}})_{k,l}} \qquad (1)$$

where $\mathbf{J}_0$ is the measured input frame and $\otimes$ denotes 2D convolution that is implemented through Fast Fourier Transform
(FFT) in the astropy Python library. Three iterations were used after finding that a greater number did not significantly alter the
correction results.

The stray light correction was relatively small for both MethaneAIR bands, approximately 2% of total detected light. A
comparison of the slit images on the $CH_4$ and $O_2$ FPAs before and after applying the stray light correction is shown in Figure 7.
The slit images appear in sharper contrast with the noise floor after correction, and the spectral stray light beyond the 15-pixel
window is substantially reduced.

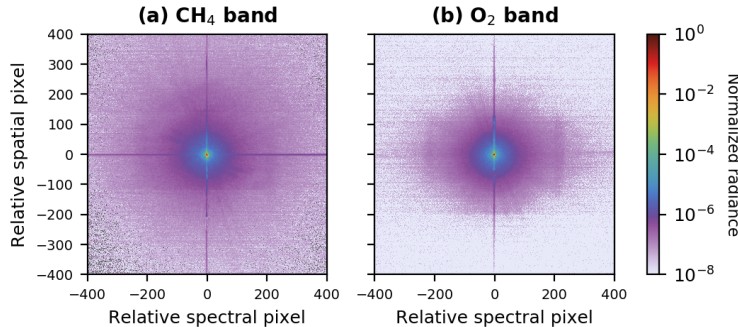

**Figure 6.** Median stray light kernels for (a) $CH_4$ and (b) $O_2$ bands. Multiple merged frames were interpolated to a common spatial/spectral
pixel grid before taking the median of all frames to produce the kernels for use in the stray light correction algorithm.

## 5 Construction of ISRF

### 5.1 Oversampling ISSF

The laser wavelength scans shown in Figure 5 yield a series of instrument spectral spread functions (ISSFs) positioned around
selected central wavelengths for each spatial pixel in each band. Since each ISSF corresponds to a wavelength, there is theoret-





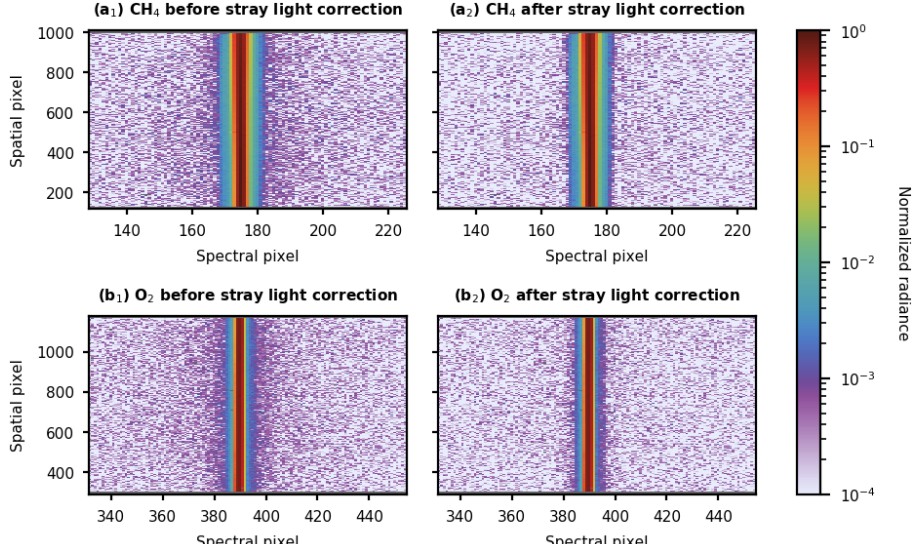

**Figure 7.** Slit images on the FPA, as given by normalized radiance before ($a_1$, $b_1$) and after ($a_2$, $b_2$) applying stray light correction. After correction, the slit image at each row (i.e., spectral pixels at a given spatial pixel) is effectively an ISSF as discussed in Section 5.

ically an infinite number of ISSFs. In contrast, there is a single ISRF for each spectral pixel, defining the response of that pixel

to light of different wavelengths. The relationship between the ISSF and ISRF is depicted in Figure 8a. Each ISSF extends over multiple spectral pixels and is comprised of samples from the ISRFs of these spectral pixels. It is assumed that the ISRF does not vary significantly over a small sample of spectral pixels. Therefore, the ISSF can be viewed as a sparsely sampled version of a representative ISRF (van Hees et al., 2018). By measuring a series of ISSFs with small wavelength steps surrounding a central wavelength and manipulating the frames to align, an oversampled ISSF can be constructed. As shown in Figure 8b for

a given central wavelength and spatial pixel, the oversampled ISSF, constructed from rows in Figure 8a, is the mirror image of the oversampled ISRF, constructed from columns in Figure 8a. We determine the ISRF by oversampling the ISSFs because it enables better corrections of laser power and wavelength fluctuation.

An iterative approach is used to construct the oversampled ISSF, starting with a series of individual ISSFs obtained by stepping the laser at 0.005 nm increments ±0.1 nm from a given central wavelength. Figures 9a and 10a exemplify a set of

measured ISSFs at one central wavelength and spatial pixel in each band. For each laser wavelength step, the center of mass and total mass of the corresponding ISSF are calculated. The centers of mass and laser wavelengths are used in an orthogonal linear regression to obtain a spectral pixel-wavelength registration function. To assemble the first oversampled ISSF, individual ISSFs are shifted horizontally by first subtracting the calculated centers of mass from the originally defined spectral pixels at each laser step. The spectral pixel center registered at the central wavelength is then added back to all ISSFs to shift the

aligned frames to the appropriate spectral pixel position. Each ISSF is also divided by its total mass to normalize the functions vertically and account for the laser power fluctuation. Figures 9b and 10b show the resulting oversampled ISSF for a specified central wavelength and spatial pixel.



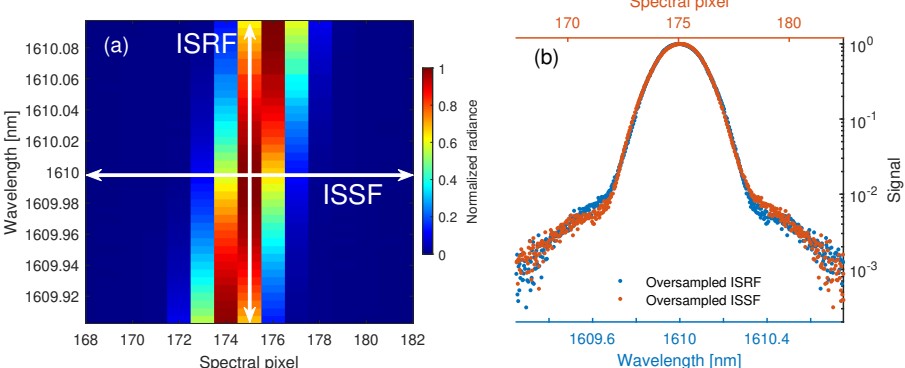

**Figure 8.** (a) Illustration of ISRFs and ISSFs near 1610 nm through the spectral responses of the row that corresponds to spatial pixel 300 of the CH$_4$ FPA. The laser was scanned from 1609.9 nm to 1610.1 nm with a step size of 0.005 nm. (b) Oversampled ISSF and ISRF. The horizontal coordinate of spectral pixel (for ISSF) is aligned with wavelength (for ISRF). The oversampled ISSF will match the ISRF if the profile shown is flipped (not shown here) and the spectral pixel coordinate is projected to wavelength space.

The oversampled ISSFs are refined by honing the shifting and scaling constants for the ISSFs at every laser wavelength step. The center of mass and total mass of each ISSF is updated by fitting a horizontal shift and vertical scale with the oversampled

ISSF constructed in the previous step. The shift and scale previously calculated are used as the initial values in this nonlinear fitting. The center spectral pixel corresponding to the central wavelength of interest is also adjusted from an updated linear fit between the new centers of mass and the set wavelength. The shifts and scaling are applied as before to assemble the improved oversampled ISSFs seen in Figures 9c and 10c. This process is repeated, though improvements after the second iteration are relatively small. Three iterations are shown for demonstration, but a total of four iterations were conducted. The spectral pixel

centers from the final iteration are saved for analysis of wavelength registration (Section 6).

After this iterative process of shifting and scaling, there is an oversampled ISSF at approximately 860 spatial positions for each of the central wavelengths in each band. In order to convert the oversampled ISSF to ISRF, the profile is flipped about its center of mass, and the horizontal coordinate is mapped from spectral pixel space to wavelength space using the wavelength registration curve fitted in the last iteration. The ISRFs are then linearly interpolated to a common wavelength grid defined

from relative wavelength -0.75 to 0.75 in 0.005 nm intervals.

### 5.2 Smoothed ISRF results

The ISRFs constructed from the oversampled ISSF data are noisy at the tails, as seen in Figure 11. Structures in the tails are inconsistent across spatial pixels and central wavelengths, so it is beneficial to smooth out these random features while preserving the ISRF shape at the core. Various analytical functions were tested to fit the ISRFs, including the TROPOMI ISRF

model described in van Hees et al. (2018), but they cannot provide sufficient fitting accuracy across all measurement positions in both MethaneAIR bands. Instead, a Savitzky-Golay filter is implemented, which fits a local polynomial to a subset of data





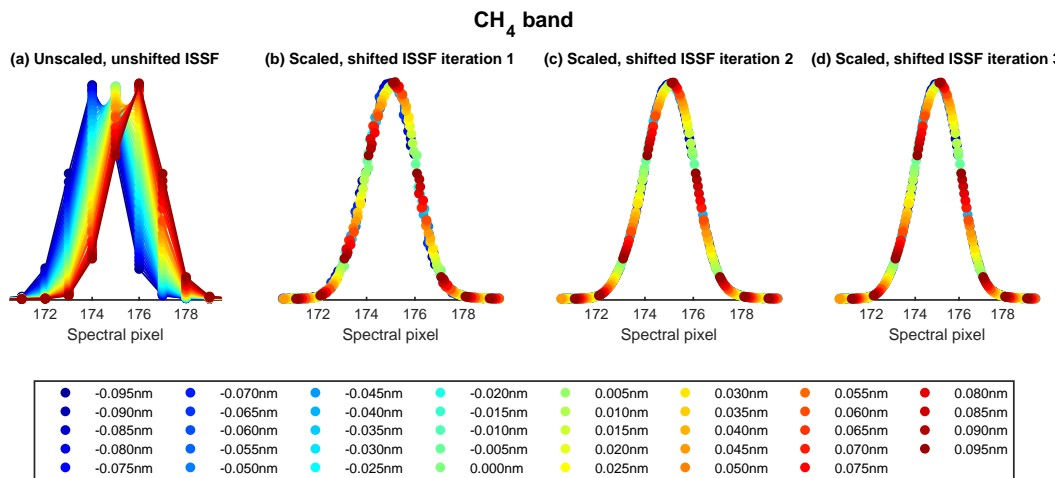

**Figure 9.** Demonstration of ISSF oversampling for the CH$_4$ band using a central wavelength of 1610 nm and spatial pixel 500. The laser was scanned in 0.005 nm steps over a range corresponding to the central wavelength $\pm$ 0.095 nm. The resulting series of individual ISSFs (a) are then shifted and scaled to produce a single oversampled ISSF (b). Successive iterations of shifting and scaling (c, d) are performed to construct smoother oversampled ISSFs, which can then be mapped from pixel space to wavelength space for ISRF analysis.

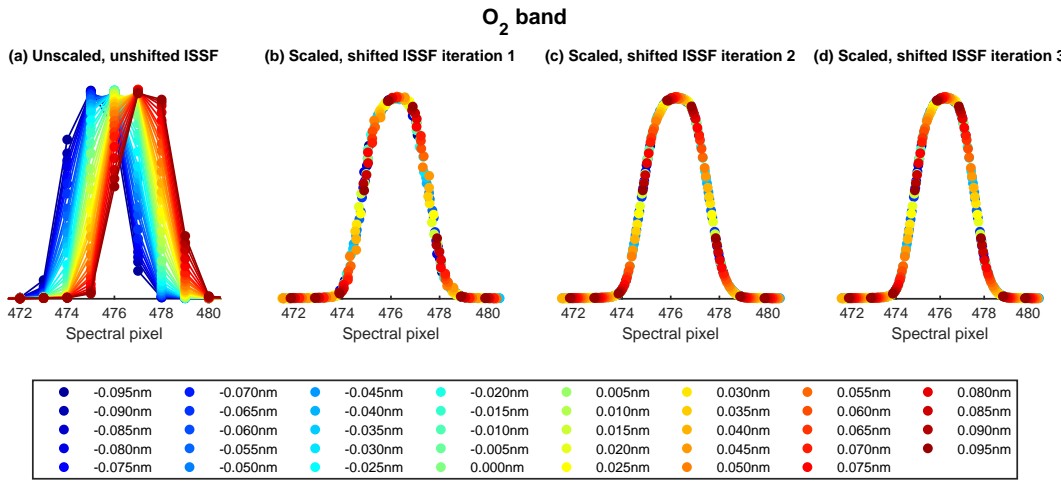

**Figure 10.** Demonstration of ISSF oversampling for the O$_2$ band at a central wavelength of 1275 nm and spatial pixel 500. ISSF processing is as described in Figure 9.





in a moving window (Savitzky and Golay, 1964). A filter of order 3 with a window length of 40 points on either side of the the central point is used, i.e., a 3.40.40 filter. The Savitzky-Golay filter was found to effectively avoid peak flattening and provide superior processing speed compared to other filters (e.g., penalized spline and robust lowess smoothing). Applying the filter once smoothed the tails fairly well, as demonstrated by the red lines in Figure 11. Still, there is room for improvement after the first pass, particularly in the $O_2$ band. In order to achieve a smoother result, an iterative version of the Savitzy-Golay filter is devised. This filter works by calculating the residuals between the logs of the raw data and the smoothed lines after an initial application of the filter. At locations outside of the core where the residuals are higher than a specified threshold, the ISRF data points are replaced by the filtered result. The same filter is then applied again to the updated set of ISRF data, and residuals are again calculated. With each iteration, the residual threshold for replacement is decreased. The numbers of iterations used for the $CH_4$ and $O_2$ bands are five and six, respectively. The result of the iterative filter shows fewer defined features in the tails, as shown in Figure 11. Values in the smoothed ISRF beyond $\pm$ 7.5 pixels from the center are set equal to zero since these values should be taken care of by the spectral stray light correction as described in Section 4. The ISRF is then normalized so that it integrates to unity.

After examining the variation of the filtered ISRF across all spatial and spectral pixels, it was evident that additional smoothing was required for a small number of pixels (74 pixels in the $CH_4$ band, and 87 pixels in the $O_2$ band). By nature, the ISRF shape should vary smoothly between spatial and spectral pixels. However, some pixels exhibited anomalous behavior by way of sharp contrasts with neighboring pixels. The performance of these atypical pixels was irregular enough to be noticeable in the ISRF results, but not poor enough to be flagged in the bad pixel map. To remove the effects of these remaining anomalous pixels, a median filter was first applied to all ISRFs, which are assembled to a table defined in spatial, spectral, and relative wavelength dimensions. The root-mean-square error (RMSE) between the original ISRF table and the median filtered ISRF table was calculated to define outliers with RMSE greater than three standard deviations from the mean for each central wavelength. Then, only ISRFs at outlier locations were replaced with the median filtered version. Due to the higher noise levels accompanying the decrease in QE at higher wavelengths in the $CH_4$ band, pixels at 1670 nm were not included in the replacement. Exceptions were also made at specific spatial pixel indices in both bands where real slit shape characteristics were seen to cause significant irregular features, which is discussed in greater detail with the wavelength registration (Section 6).

Examples of the smoothed ISRF shapes after Savitzky-Golay filtering and outlier smoothing are given in Figures 12 and 13. Figure 14 displays the ISRF full width at 20%, 50%, and 80% of peak height, conveying the variation in ISRF shapes in both bands across the FPA. As shown by the figures, the ISRF is often asymmetric at both the core and the tails. The ISRF is broader and more triangular in the $CH_4$ band compared to the $O_2$ band. Additionally, the shape tends to grow wider with increasing spatial and spectral indices in the $CH_4$ band, as seen in Figure 14a-c. In contrast, the $O_2$ band exhibits much less variation, indicated by the relatively narrow color ranges in Figure 14d-f. The smoothed ISRFs are saved to produce a three dimensional lookup table defined for each illuminated spatial pixel, each central wavelength, and a relative wavelength grid from -0.75 nm to 0.75 nm with 0.005 nm steps for each ISRF. The use of only central wavelength positions in the table rather than all spectral pixels is enabled by the smooth spectral variation of the ISRF; all spatial positions are defined due to more irregular spatial variation.





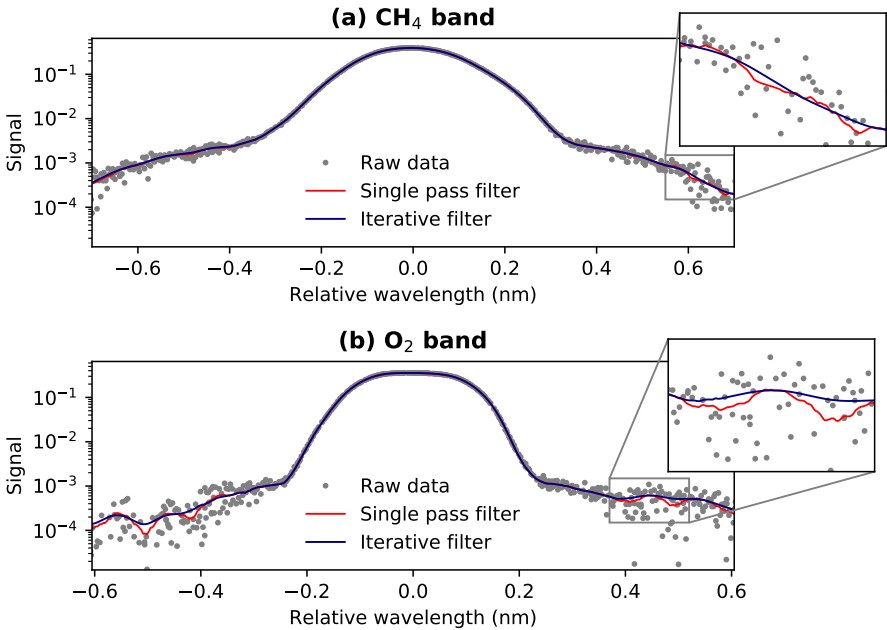

**Figure 11.** Demonstration of the iterative Savitzky-Golay filter used to smooth ISRF measurement data in the $CH_4$ (a) and $O_2$ (b) bands. One application of the filter, shown in red, was fairly effective, especially for the $CH_4$ band. Successive iterations applied to the residuals at the tails provided additional smoothing while preserving the ISRF shape at the core, as indicated by the blue line.

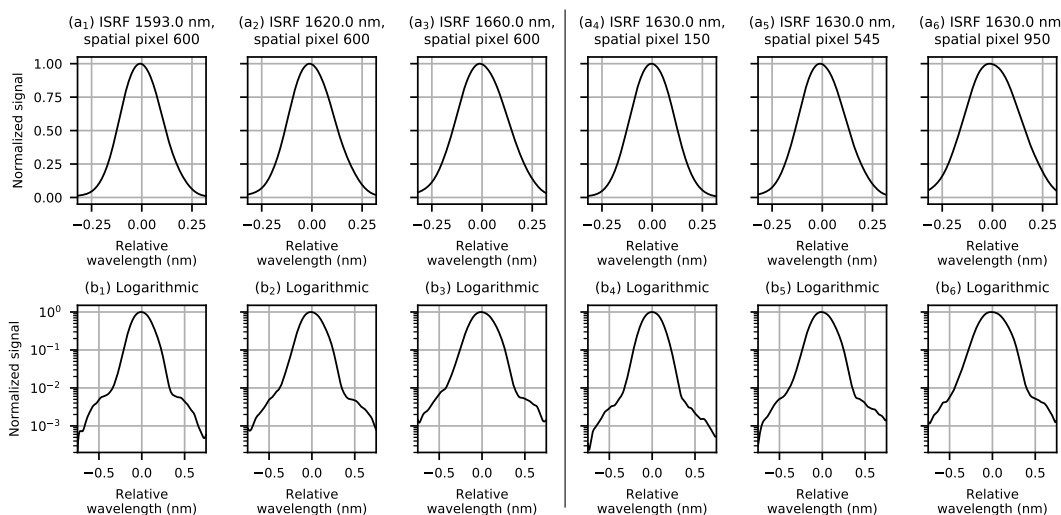

**Figure 12.** ISRF shapes at various positions for the $CH_4$ band. The first three panels ($a_1$-$a_3$) demonstrate the variation in shape for different central wavelengths at a constant spatial position, while the rightmost three panels ($a_4$-$a_6$) hold central wavelength constant to show ISRF changes across spatial pixels. The bottom row of panels ($b_1$-$b_6$) displays the same data on a logarithmic scale and wider relative wavelength range. The ISRFs are normalized so the maximum is unity.





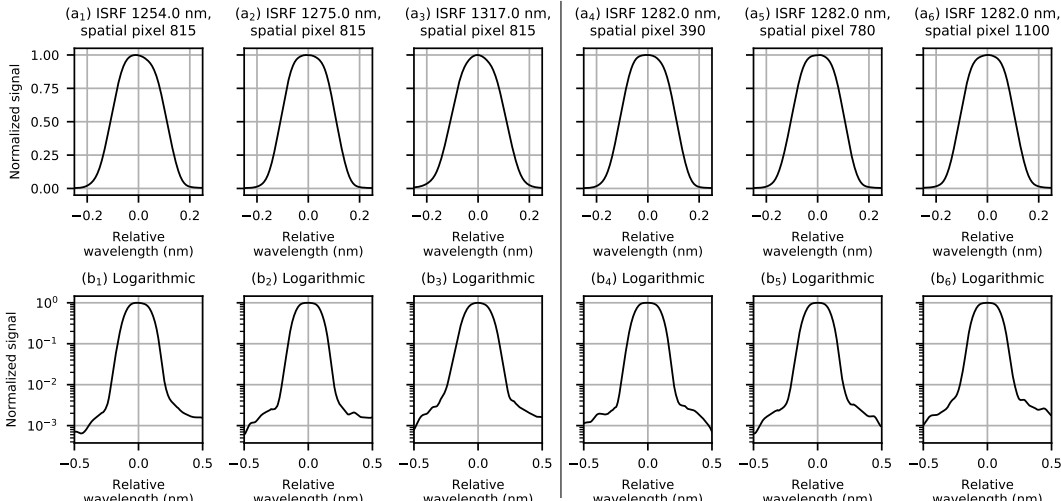

**Figure 13.** Similar to Figure 12 but for the $O_2$ band.

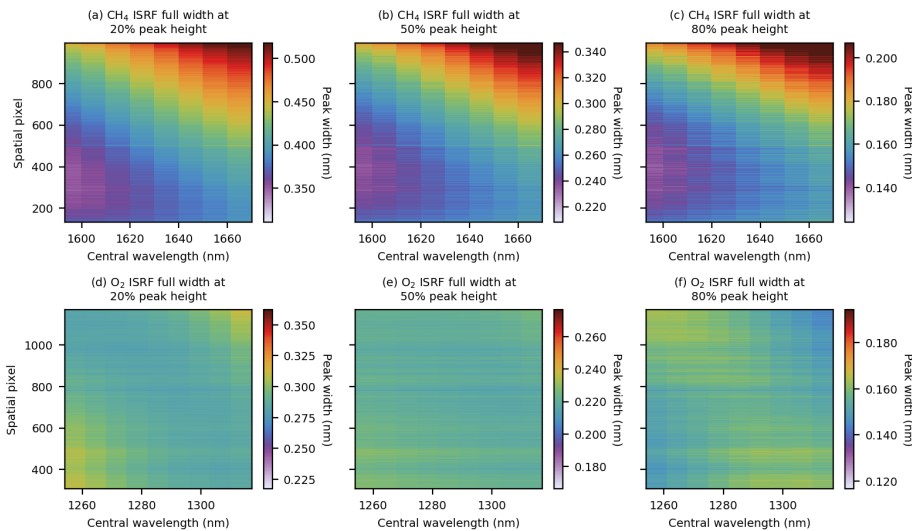

**Figure 14.** Full width of the smoothed ISRF at 20% (a,d), 50% (b,e), and 80% (c,f) of maximum peak height for all spatial and spectral pixels. The top three panels correspond to the $CH_4$ band, which shows a general broadening of the ISRF with increasing wavelength and spatial pixel index. In contrast, the $O_2$ band ISRF is more homogeneous across different spatial and spectral pixels, as reinforced by the relatively narrower scales for the bottom three panels. The color limit in each panel is fixed at $\pm 25\%$ from the FPA-mean value.





## 6    Wavelength registration

The ISRF construction process as previously described resulted in spectral pixel centers that correspond to the laser central wavelengths labeled in Figure 5. Those spectral pixel-wavelength relationships are determined with high accuracy for all

illuminated spatial pixels. It is possible to derive the wavelength registration function for each spatial pixel by independently fitting the spectral pixel centers vs. laser central wavelengths. However, we noticed some outliers that are caused by either inadequate filtering of bad pixels or the deficiency in the ISRF after a significant number of pixels are removed as bad pixels, as shown in Figure 15. To prevent the impact of those localized outliers from propagating to the wavelength calibration curves that cover the full spectral range, we apply an additional smoothing to the spectral pixel centers as described in the following.

For each central wavelength, the median is removed from the spectral pixel centers of all spatial pixels, and the resultant relative spectral pixel values are highly consistent for all central wavelengths, as shown by the dots in Figure 15. The fine scale structures in the spatial dimension likely originate from irregularities of the slit along its length. Such structures are most easily seen near spatial pixel 505 in the $CH_4$ band and spatial pixel 780 in the $O_2$ band. Those structures are also observable in the ISRF widths shown in Figure 14. The medians along the wavelength dimension are then taken from the combined relative

spectral pixel values of all wavelengths, represented by the black lines in Figure 15. These series of median values remain largely unaffected by the random noise or outliers at individual wavelengths while preserving the structures that are common to all wavelengths. Finally, a linear fit is made between those median values and the spectral pixel center values for each center wavelength, and the predicted spectral pixel center values, which are smooth and free of outlier points, are used in the final wavelength calibration.

In both bands, a polynomial fit is applied to the smoothed spectral pixel centers as a function of wavelength. This is necessary to map spectral pixel to wavelength at locations between the measured points. As shown by the bottom panels of Figure 16, the residuals for various polynomial degrees are quite similar, and less than approximately 0.02 spectral pixel. For clarity, only first through fourth order polynomial residuals are plotted, but higher orders, up to and including seventh, were tested. A first order polynomial was selected as the optimal model for both bands, in accordance with the Akaike information criterion (AIC)

(Akaike, 1974) and Bayesian information criterion (BIC) (Schwarz, 1978). The linear fit between spectral pixel index and wavelength is shown in the top panels of Figure 16.

## 7    Flight spectra demonstration

Here we evaluate the performance of the on-ground MethaneAIR calibration using radiance spectra from the first instrument flight over a clean region of the Colorado Front Range, using the optimal-estimation-based (Rodgers, 2000) retrieval algorithm

being developed for MethaneAIR/MethaneSAT (Chan Miller et al., 2018). Further detailed description of the algorithm will be provided in future publications on MethaneAIR retrieval. The Level 0 detector signals are converted to Level 1b radiance spectra through dark current subtraction, bad pixel removal, radiometric calibration, and stray light correction in a similar way as the ISRF calibration data. In addition, the wavelength-dependent viewport window transmittance is corrected. Fits for spectra in the $O_2$ and $CH_4$ bands are used for cloud filtering and $CH_4/CO_2$ proxy retrieval, respectively. The algorithm settings





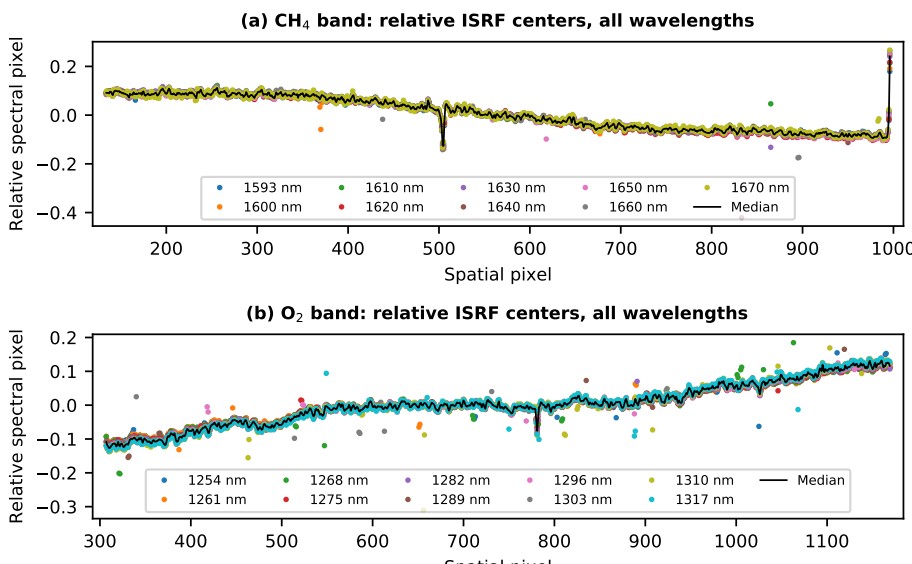

**Figure 15.** For each wavelength, the spectral pixel center at every spatial pixel is shifted to align all wavelengths for a given band. The median of the combined data is taken (black line), resulting in a smoothed version of the central spectral pixel indices for all spatial pixels. Variations common to all wavelengths, such as the feature near spatial pixel 505 in the $CH_4$ band (a) are preserved in the smoothing. These are real features, likely due to irregularities in the slit width.

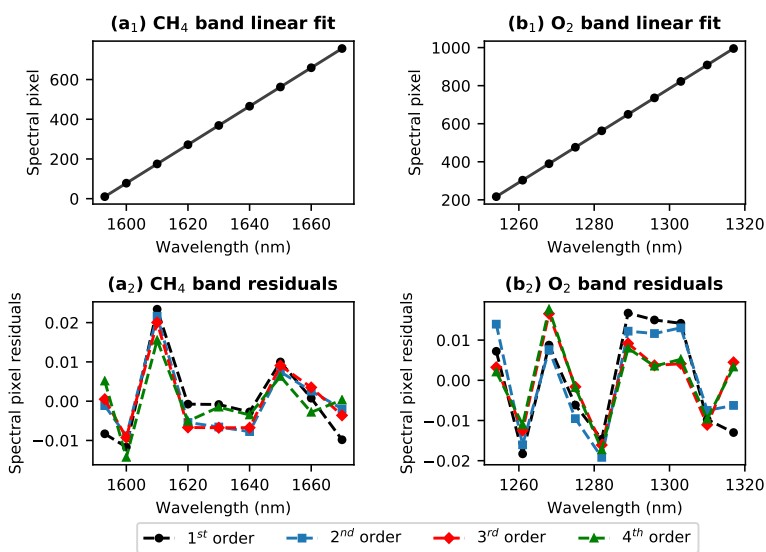

**Figure 16.** Wavelength registration fitting results for $CH_4$ (a) and $O_2$ (b) bands, demonstrated for spatial pixel 600. Top panels show a linear fit to the data, which was determined to be the optimal polynomial order by AIC and BIC. Residuals from first through fourth order polynomial fits are given in the bottom panels.





are summarized in Table 2. Scattering is neglected in both retrievals; A reasonable assumption since Rayleigh scattering is negligible, and aerosol loadings during the flight were low (observed 1640 nm aerosol optical depths were < 0.01 at the AERONET NEON-CPER site, close to the flight path).

**Table 2.** MethaneAIR Level 2 algorithm fit settings

| State Vector Element | A Priori | Error | O$_2$ Band[1] | CH$_4$ Band[2] |
|---|---|---|---|---|
| CH$_4$ Profile | TCCON GGG2020[3] | Altitude-correlated covariance[4] | ✗ | ✓ |
| CO$_2$ Column | TCCON GGG2020 | 12 ppmv | ✗ | ✓ |
| H$_2$O Column | GEOS-FP[5] | 0.02 v/v | ✓ | ✓ |
| O$_2$ CIA pseudo absorber | 0.21 v/v | 15% | ✓ | ✗ |
| Temperature Profile Shift | GEOS-FP | 5 K | ✓ | ✓ |
| Surface Pressure | GEOS-FP | 4 hPa | ✓ | ✓ |
| Albedo | Derived from observation | 100% | 5th order | 3rd order[6] |
| Wavelength Offset | 0.0 nm | 0.01 nm | ✓ | ✓ |
| ISRF squeeze | 1.0 | 0.2 | Optional | Optional[6] |

[1] Fit window 1249.2–1287.8 nm (O$_2$)

[2] Two fit windows: 1595–1610 nm. (CO$_2$), 1629–1654nm (CH$_4$)

[3] Laughner et al. (2020)

[4] 6-km vertical length scale

[5] Knowland et al. (2020)

[7] Different for CO$_2$/CH$_4$ window

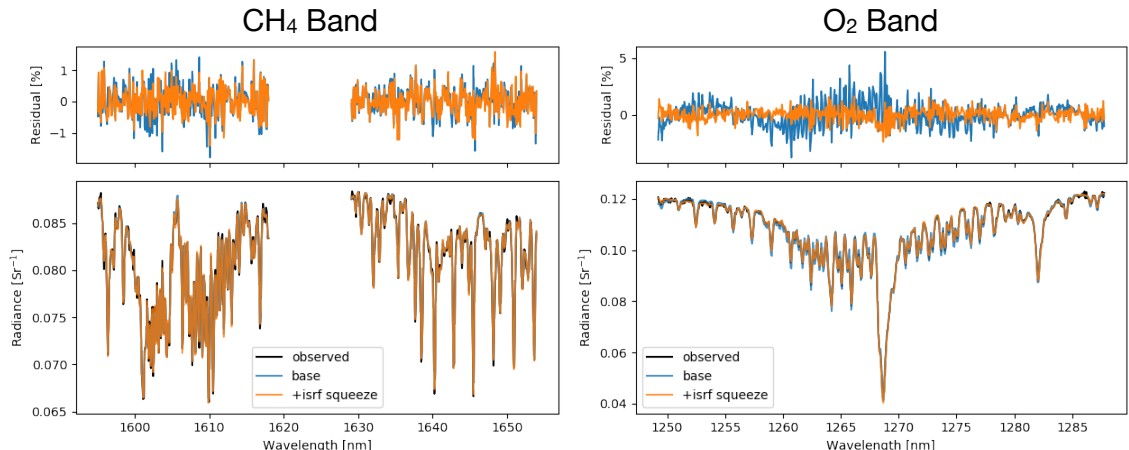

**Figure 17.** Spectral fits from MethaneAIR Research Flight 1 (11/8/2019, 18:36 UTC) using the MethaneAIR optimal-estimation algorithm. Spectra are 10 second along-track aggregates for spatial pixel 600 (approximate center of detector). Blue color indicates the fit and residual using the laboratory calibrated ISRF look-up tables, and orange color indicates the fit and residual with an ISRF squeeze parameter.





Figure 17 shows spectral fits for each band for an across track position at the center of the detector. The spectra constitute a 10 s along-track aggregate of frames, taken when the flight was at cruise altitude ($\sim$12 km). Time aggregation was performed to

boost signal-to-noise and mitigate the impact of inhomogeneous slit illumination on the ISRF. Applying the nominal calibration derived in this paper is shown by the blue lines, leading to fit residual RMSE of 1.12% and 0.52% in the $O_2$ and $CH_4$ bands, respectively.

The large difference in the residuals between instruments and simulations could be due to a change in the detector focus from on-ground to in-flight especially for the $O_2$ spectrometer. This may arise from a difference from the lab and flight environments, such as

such as a change in the temperature of the optical bench or mechanical stress of the instrument. To first order, these changes manifest as a change in ISRF width, which can be modeled by scaling the wavelength grid ($\lambda$) of the tabulated ISRF ($\Gamma_{TAB}(\lambda)$) (Sun et al., 2017) via squeeze parameter ($x_{sqz}$):

$$\Gamma(\lambda) = \Gamma_{TAB}(x_{sqz}\lambda) \tag{2}$$

The orange lines in Figure 17 show the improvement in spectral fits after including $x_{sqz}$ in the retrieval state vector. The fitted

$x_{sqz}$ for the $O_2$/$CH_4$ bands for those particular across-track positions are 0.865 and 1.055, representing a broadening/narrowing of the ISRF, respectively. Accounting for changes to the ISRF width yields comparable fit RMSE for both channels (0.6% for $O_2$ and 0.45% for $CH_4$). Application of the ISRF squeezing improves the fitting quality in other across-track positions similarly. This indicates that the systematic difference between in-flight and on-ground calibration of ISRF needs to be accounted for in the retrieval algorithm.

**8 Conclusions**

This paper focuses on the spectral calibration of MethaneAIR including stray light correction, ISRF characterization, and wavelength calibration. The stray light was stable in both bands, allowing for the use of a position-independent median kernel in the correction algorithm based on Van Cittert deconvolution. The correction was rather minor since stray light accounted for only a small fraction of the total detected light.

The ISRF was determined by first oversampling the ISSF around roughly ten central wavelengths in each band. Each oversampled ISSF was reflected about its center of mass and projected to a fine wavelength grid to transform the profile into an ISRF. This ISSF approach, which allows for more precise correction of laser power/wavelength fluctuation, approximates the true ISRF better than direct ISRF measurements. The ISRFs were further processed by applying an iterative Savitzky-Golay filter to smooth high-frequency noise at the tails. Final ISRFs were saved to a lookup table for use in the retrieval algorithm

since the shapes could not be satisfactorily modeled by an analytical function. The observed shape of the ISRF peak was more triangular in the $CH_4$ band compared to the $O_2$ band. The ISRF shape in the $CH_4$ band varied considerably more than in the $O_2$ band in both spatial and spectral dimensions. This increased variability in the $CH_4$ band may have been due to optical influences from the internal alignment of the instrument. In contrast, the $O_2$ ISRF full width at half-maximum (FWHM) was dominated by the slit width, which is essentially constant. Analysis of the wavelength-spectral pixel relationship found that a

linear wavelength calibration is sufficient after reducing individual noise contributions.





The performance of the on-ground MethaneAIR spectral calibration was demonstrated using radiance spectra retrieved from an instrument flight over the Colorado Front Range. Fitting the base calibration from the ISRF lookup table to the spectra resulted in larger residual RMSE for the $O_2$ band than the $CH_4$ band, which was presumably caused by a change in detector focus in flight. Slight differences in environmental conditions between lab and flight situations could contribute to this change,

embodied by an adjustment in the ISRF width. Scaling the wavelength grid of the tabulated ISRF by a constant parameter improves the spectral fit in both bands. This squeeze factor indicated a broadening of the ISRF in the $O_2$ band and a narrowing in the $CH_4$ band.

The general calibration framework as well as specific insights gained from MethaneAIR may help to advance the future spectral calibration of MethaneSAT. Stray light measurement data showed that the partial illumination of the slit width distorted

the ISRF. In future stray light measurements, the pinhole will be replaced with a thin slit in order to fully illuminate the width of the spectrometer slit. Similarly, the MethaneAIR ISRF construction process and results can be used to inform the necessary ISSF measurement extent for MethaneSAT. Measurements at ten or so central wavelengths appears to be adequate, given that the ISRF varies smoothly in the spectral dimension. However, the degree of spatial variation seen in the MethaneAIR ISRF suggests that it is important to assess all pixels in the spatial dimension. Application of the calibration to real flight

data demonstrated the possibility that the ISRF width may change between on-ground calibration and in-flight or on-orbit conditions; however, this may be compensated for by including a scaling parameter in the retrieval algorithm.

*Competing interests.* The authors declare that they have no conflict of interest.

*Acknowledgements.* We acknowledge MethaneSAT, LLC and the Environmental Defense Fund to support MethaneSAT and MethaneAIR

Science and Algorithms at University at Buffalo and the Smithsonian Astrophysical Observatory. The MethaneAIR flight program was supported by the National Science Foundation, grant 1856426 to Harvard University.



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
