# Peer review of "Spectral Calibration of the MethaneAIR Instrument"

_Atmospheric Measurement Techniques, 2020_

## Referee Comment (RC1)

Referee report

The report is well written, extensive and covers a topic that is relevant to the scientific community. I recommend it for publication with some revisions.

Major points

- I would like to see more relevance for MethaneAIR w.r.t. MethaneSAT. Is the method presented here is only briefly revisited in the last few paragraphs of the conclusions. More and/or an expanded discussion on this point likely improves the quality of the paper.
- Section 2: Although less important than the ISRF, the PSF size should be mentioned.
- Section 3: I am missing information on source (i.e. laser) stability for the measurements as well as expected noise levels.
- Figures: Some figures are relatively small, with important details being referenced at times difficult to discern. Enlarging some figures (notably Fig. 4, 5 ,6,7, 12, 13, 14) greatly enhances their effect.
- Figure 6: Straylight kernels are often four times bigger than the detector itself to account for far-field. See e.g. Fig. 11a in Tol et al., 2018. This is not given here. Even if these far-field straylight are within the noise, this is a key part of the kernel. Please give the full kernel.
- Figure 14: For the total 'smoothed' ISRF there are discrete transitions. It appears there is no truly smooth ISRF. Why was this done? Please explain.
- Section 7: Only the ISRF was validated in flight. Although in that derivation a straylight correction was done, this is not presented. Was there some activity on straylight validation?

Minor points
- Line 2: …anthropogenic CH4 point sources…
- Line 2 (as well as later): Define scale of 'basin'
- Line 22 : minor, but GOSAT is a satellite, while TROPOMI is an instrument. They are treated equally here. Sentinel-5P is the satellite carrying TROPOMI.
- Line 30/31: Define 'intermediate' scales
- Line 32/33: I assume the swath is 200 km, not 200 km2?
- Line 36: Please motivate the choice for the 1.27 O2 band more (e.g. explain recent advances, why proximity is better)
- Line 64: Remove 'only'
- Line 64 - 69 : Only TROPOMI is mentioned here. Are there other stray light treatments, e.g. GOSAT, GOSAT2, that are worth mentioning?
- Line 71 or 80: similar(ity) – At times it must be guessed how similar or dissimilar MethaneSAT is from MethaneAIR (see major point). Please quantify as much as possible.
- Line 81: quantify swath width in text. One important part is the swath *angle* difference between MethaneSAT and MethaneAIR.
- Line 86: sub-pixel *spectral* smile
- Line 83: Figure 5 referenced before Figures 2,3 and 4.
- Line 118: I do not understand the motivation to force the intercept to be zero.

- Line 136-139: What is the relevance of these statements?
- Line 149: Quantify the decrease in QE
- Figure 5 : Please mark which laser responses were done with different power to account for the decrease in QE
- Line 186 : Is the 2 % for both? Please quantify for either spectrometer.
- Figure 10: Are these normalized? Please give Y-axis.
- Line 242: Please give ISRF beyond 7.5 pixels in nm as well.
- Line 246: Confusing. These are individual pixels within the full spatial illumination of a laser? And are these not correlated with the bad pixels mentioned earlier?
- Section 7 : Please give dates and lengths of flight. Date of flight can only be read from caption of Fig. 17.
- Line 303: Why was only a single across track position done? This could be repeated for similar across track positions at other points during the flight. Are results similar?
- Line 308 : Difference is attributed to temperature changes due to environment. Could the temperature difference be quantified?

---

## Author Comment (AC1)

Response to Referee #1:

We appreciate the very helpful feedback from the referee. The referee's comments are listed in *italics*, followed by our response in blue. New/modified text in the manuscript is in **bold**.

*Major points*:

*1. I would like to see more relevance for MethaneAIR w.r.t. MethaneSAT. Is the method presented here is only briefly revisited in the last few paragraphs of the conclusions. More and/or an expanded discussion on this point likely improves the quality of the paper.*

Information on the similarities and differences between the MethaneAIR and MethaneSAT instruments has been added to Section 2, at line 80 of the original manuscript.

**"The airborne instrument provides similar spectroscopy to MethaneSAT with higher spatial resolution, although with a significantly smaller swath width (5.05 km at a flight altitude of 12 km for MethaneAIR vs. ~260 km for MethaneSAT) due to the difference in operating altitude. The MethaneAIR point spread function (PSF) is roughly 2.5 pixels wide across-track, estimated from the spatial stray light data. The swath angles of MethaneAIR and MethaneSAT are similar, at 23.7° and 21.3°, respectively. While MethaneSAT has a larger FPA of 2048 × 2048 pixels compared to the 1024 × 1280 pixels of MethaneAIR, the spectral range of the satellite instrument is reduced due to illumination of only 1000 spectral pixels at most. The spectral resolution of MethaneSAT, however, is 20%-30% higher than that of MethaneAIR. One of the most significant differences between the instruments is the detector material; the satellite detector is Mercury-Cadmium-Telluride (MCT), unlike that of MethaneAIR."**

The stray light correction for MethaneSAT is under development and will be similar to (but more sophisticated than) the method presented for MethaneAIR. The following sentence has been added to line 151 to tie in MethaneSAT:

**"This method will also be applied to MethaneSAT, incorporating lessons learned from MethaneAIR."**

The following sentence has been added to line 225 of the original manuscript regarding the connection between MethaneAIR and MethaneSAT for ISRF characterization:

**"The spectral calibration of MethaneSAT will follow and build upon the method of ISRF determination presented here for MethaneAIR."**

In addition, the retrieval algorithm used to evaluate the calibration in Section 7 is being developed for both MethaneAIR and MethaneSAT, as pointed out in the first sentence of Section 7.

*2. Section 2: Although less important than the ISRF, the PSF size should be mentioned.*

The PSF may be estimated from the spatial stray light data to be roughly 2.5 pixels wide across-track. This approximation is limited given that data were collected at only three spatial positions. The following sentence is added to the first paragraph in Section 2:

"**The MethaneAIR point spread function (PSF) is roughly 2.5 pixels wide across-track, estimated from the spatial stray light data.**"

*3. Section 3: I am missing information on source (i.e. laser) stability for the measurements as well as expected noise levels.*

The following sentences were inserted at line 127 of the original manuscript to quantify the laser stability:

"**The wavelength stability of the $CH_4$ laser is $\pm 5$ pm, and the optical power is +1.2%/-0.9%. For the $O_2$ laser, the wavelength stability is +5/-6 pm, and the optical power is +1.1%/-0.5%.**"

*4. Figures: Some figures are relatively small, with important details being referenced at times difficult to discern. Enlarging some figures (notably Fig. 4, 5 ,6,7, 12, 13, 14) greatly enhances their effect.*

The referenced figures have been enlarged.

*5. Figure 6: Straylight kernels are often four times bigger than the detector itself to account for far-field. See e.g. Fig. 11a in Tol et al., 2018. This is not given here. Even if these far-field straylight are within the noise, this is a key part of the kernel. Please give the full kernel.*

We don't have comparable spatial coverage on the FPA to TROPOMI (3 swath angles in MethaneAIR vs. 99 swath angles in TROPOMI SWIR), so we have to focus on the relative near field, where more than one coverage can be achieved. The spectral coverage in the $CH_4$ band is further limited by the QE drop beyond 1656 nm, where the noise is too large. The following statement is added to the end of the second paragraph in Section 4 to clarify this point:

"**This process ideally gives a stray light kernel that is up to four times larger than the FPA. However, due to the limited spatial swath angles, the region near the spatial dimension edges are often subject to excessive noise. The spectral coverage in the $CH_4$ band is further**

**limited by the QE drop towards the long wavelength. As such, we limit the stray light kernel to be within +/-400 pixels for both spatial and spectral dimensions."**

*6. Figure 14: For the total 'smoothed' ISRF there are discrete transitions. It appears there is no truly smooth ISRF. Why was this done? Please explain.*

The transitions correspond to the discrete wavelengths where ISRF measurements were collected. Figure 14 has been updated using Gouraud shading in order to show the smooth ISRF variation across the FPA. The updated Figure 14 and caption is included here:

"

[Figure]

**Figure 14. Full width of the smoothed ISRF at 20% (a,d), 50% (b,e), and 80% (c,f) of maximum peak height for all spatial and spectral pixels. Gouraud shading is applied to render smooth ISRF variation across the FPA. Wavelength labels have been projected to the abscissa in order to provide more context compared to the spectral pixel index. The top three panels correspond to the CH₄ band, which shows a general broadening of the ISRF with increasing wavelength and spatial pixel index. In contrast, the O₂ band ISRF is more homogeneous across different spatial and spectral pixels, as reinforced by the relatively narrower scales for the bottom three panels. The color limit in each panel is fixed at +/- 25% from the FPA-mean value."**

*7. Section 7: Only the ISRF was validated in flight. Although in that derivation a straylight correction was done, this is not presented. Was there some activity on straylight validation?*

The stray light correction is demonstrated for laboratory ISRF measures as shown in Figure 7. Currently we don't have sufficient resources for in-flight stray light validation given the limited flight hours. Potential validation efforts include retrieval sensitivity test over high-contrast scenes and comparison with ground-based measurements (e.g., from EM-27), but those are not deemed to be within the scope of this paper.

*Minor points:*
*1. Line 2: …anthropogenic CH4 point sources…*

The words "$CH_4$" and "point" have been swapped, as suggested.

*2. Line 2 (as well as later): Define scale of 'basin'*

The first sentence of the abstract has been revised.

**"MethaneAIR is the airborne simulator of MethaneSAT, an area-mapping satellite currently under development with the goal of locating and quantifying large anthropogenic $CH_4$ point sources as well as diffuse emissions at the spatial scale of an oil and gas basin."**

Clarification has been added to the sentence referring to 'basin-scale' at line 32:

**"MethaneSAT aims to characterize oil and gas basin-scale, diffuse $CH_4$ emissions through a wide swath of 260 km, and at the same time locate and quantify large point sources within each target area that is typically $200 \times 140$ km² (Benmergui et al., 2020; MethaneSAT, LLC, 2020).**

*3. Line 22: minor, but GOSAT is a satellite, while TROPOMI is an instrument. They are treated equally here. Sentinel-5P is the satellite carrying TROPOMI.*

This sentence at line 22 has been revised to provide TROPOMI satellite information.

**"Currently, the Greenhouse gases Observing SATellite (GOSAT) and TROPOspheric Monitoring Instrument (TROPOMI) aboard the Sentinel-5P satellite collect $CH_4$ abundance data at a global scale (Yoshida et al., 2011; Hu et al., 2016)."**

*4. Line 30/31: Define 'intermediate' scales*

The use of 'intermediate scale' references the previous sentence (line 28) to be loosely defined as a scale larger than that of GHGSat-D and smaller than that of TROPOMI. The sentence at line 30/31 has been revised:

**"MethaneSAT is a push broom imaging satellite under development that is designed to operate at a scale in between that of current target-mode satellites and global mappers."**

*5. Line 32/33: I assume the swath is 200 km, not 200 km2?*

Upon re-checking, the swath at nadir is 260 km (not km$^2$). The sentence at line 32/33 in the original manuscript has been updated:

**"MethaneSAT aims to characterize oil and gas basin-scale, diffuse CH$_4$ emissions through a wide swath of 260 km and at the same time locate and quantify large point sources within each target area that is typically 200 × 140 km$^2$ (Benmergui et al., 2020; MethaneSAT, LLC, 2020).**

*6. Line 36: Please motivate the choice for the 1.27 O2 band more (e.g. explain recent advances, why proximity is better)*

The text at line 36 in the original manuscript has been revised to include more information about the O$_2$ $a^1$ $\Delta_g$ band.

**"Although the O$_2$ A band (~0.76 μm) has been commonly used for this purpose, the O$_2$ $a^1$ $\Delta_g$ band may be more advantageous. Recent advances in separating emitted airglow from backscattered light enable the use of the O$_2$ $a^1$ $\Delta_g$ band in remote sensing applications (Sun et al., 2018; Bertaux et al., 2020). Further, the close spectral proximity of the O$_2$ $a^1$ $\Delta_g$ band to the CH$_4$/CO$_2$ bands is favorable, as any differences in aerosol and cloud optical properties between the bands are reduced (Sun et al., 2018)."**

*7. Line 64: Remove 'only'*

Removed.

*8. Line 64 - 69 : Only TROPOMI is mentioned here. Are there other stray light treatments, e.g. GOSAT, GOSAT2, that are worth mentioning?*

GOSAT1/2 are scanning Fourier transform spectrometers where stray light behaves differently than imaging grating spectrometers. OCO-2/3 are similar grating spectrometers with many fewer across-track footprints (8), and the stray light is simply treated as a zero-level offset to our knowledge. The TROPOMI SWIR band is the closest reference. We have added the sentence at line 66:

**"The MethaneAIR stray light is measured and corrected for following the TROPOMI SWIR band, which is the closest reference to MethaneAIR with a wide across-track swath."**

*9. Line 71 or 80: similar(ity) – At times it must be guessed how similar or dissimilar MethaneSAT is from MethaneAIR (see major point). Please quantify as much as possible.*

As stated in response to major point #1, the sentence at lines 80-81 of the original manuscript has been revised to include swath widths.
The following text has also been added to provide detail regarding the similarities and differences between MethaneAIR and MethaneSAT.

**"The airborne instrument provides similar spectroscopy to MethaneSAT with higher spatial resolution, although with a significantly smaller swath width (5.05 km at a flight altitude of 12 km for MethaneAIR vs. ~260 km for MethaneSAT) due to the difference in operating altitude. The swath angles of MethaneAIR and MethaneSAT are similar, at 23.7° and 21.3°, respectively. While MethaneSAT has a larger FPA of 2048 × 2048 pixels compared to the 1024 × 1280 pixels of MethaneAIR, the spectral range of the satellite instrument is reduced due to illumination of only 1000 spectral pixels at most. The spectral resolution of MethaneSAT, however, is 20%-30% higher than that of MethaneAIR. One of the most significant differences between the instruments is the detector material; the satellite detector is Mercury-Cadmium-Telluride (MCT), unlike that of MethaneAIR."**

*10. Line 81: quantify swath width in text. One important part is the swath \*angle\* difference between MethaneSAT and MethaneAIR.*

The sentence at line 81 in the original manuscript has been edited to include swath widths, and an additional sentence has been added to provide the swath angles:

**"The airborne instrument provides similar spectroscopy to MethaneSAT with higher spatial resolution, although with a significantly smaller swath width (5.05 km at a flight altitude of 12 km for MethaneAIR vs. ~260 km for MethaneSAT). The swath angles of MethaneAIR and MethaneSAT are similar, at 23.7° and 21.3°, respectively."**

*11. Line 86: sub-pixel \*spectral\* smile*

The word "spectral" has been inserted into the sentence at line 86.

**"The optical design provides sub-pixel spectral smile and keystone distortion and relatively uniform focus across wavelength and field angle."**

*12. Line 83: Figure 5 referenced before Figures 2,3 and 4.*

The figure order has been rearranged such that the original Figure 5 is now Figure 2.

*13. Line 118: I do not understand the motivation to force the intercept to be zero.*

The dark current has been removed so zero digital number should correspond to zero radiance. If an intercept is included in the fitting, errors in the upper range of the calibration curve will

propagate to the intercept, potentially leading to very large relative errors in the low radiance range. Lines 116-118 are updated:

**"Flat field data were taken at exposure times of 50, 100, and 150 ms, matching the exposure times used in flight and in the ISRF calibration, and the dark frames were subtracted. This resulted in curves of spectral radiance (phot s-1 cm-2 sr-1 nm-1) vs. dark-subtracted focal plane intensity (DN s-1) for each exposure time and every active pixel. These radiometric calibration curves were fitted by fifth-order polynomials with the intercept forced to be zero. The zero intercept is necessary to guarantee zero radiance at zero DN."**

*14. Line 136-139: What is the relevance of these statements?*

This paragraph has been revised to provide more context.

**"The measurements described above are preliminary and were used primarily to develop the stray light correction algorithm. In the near future, the stray light will be measured with higher precision and an updated correction will be derived. Improvements to the measurement setup are currently underway and include automated tilt and translation stages to address many more field angles and an all-reflective collimator to avoid stray reflections from the refractive corrector plate. In addition, the pinhole will be replaced with a 100 µm slit oriented perpendicular to the spectrometer entrance slit, in order to fill the width of the spectrometer slit while providing a point source in the across-track dimension."**

*15. Line 149: Quantify the decrease in QE*

Lines 90-91 address the QE decrease. The sentence in the original manuscript has been revised to include additional data points:

**"The QE begins to roll off above this wavelength, decreasing to 0.52 at 1660 nm, then 0.26 at 1670 nm before reaching a minimum of about 0.15 at the 1680 nm end of the $CH_4$ passband."**

*16. Figure 5 : Please mark which laser responses were done with different power to account for the decrease in QE*

The sentence at lines 147-148 of the original manuscript has been revised to provide the wavelengths and laser power range, which may help to further address comment #15 above.

**"In the $CH_4$ band, the laser power was increased progressively from -3.0 dBm at wavelengths ≤1640 nm, up to +2.5 dBm at 1670 nm to compensate for the decrease in QE at longer wavelengths."**

*17. Line 186 : Is the 2 % for both? Please quantify for either spectrometer.*

Both are approximately 2%, if a $\pm 400$ spectral and spatial pixel window is defined. The sentence at line 186 has been removed, and the following sentence is added to line 177:

**"The sum of the far-field kernel is 2.4% for the CH$_4$ band and 2.1% for the O$_2$ band. This indicates that the stray light is small relative to the useful signals at each spatial and spectral position."**

*18. Figure 10: Are these normalized? Please give Y-axis.*

The first unscaled, unshifted panel is not normalized. The y-axis has been added to Figures 9 and 10.

*19. Line 242: Please give ISRF beyond 7.5 pixels in nm as well.*

Line 242 in the original manuscript has been revised, and a sentence has been added to include the quantity in nm:

**"Since values in the smoothed ISRF beyond $\pm$ 7.5 pixels from the center should be taken care of by the spectral stray light correction as described in Section 4, these values are set equal to zero. This area corresponds to pixels outside of the central wavelength $\pm$ 0.75 nm and $\pm$ 0.6 nm in the CH$_4$ and O$_2$ bands, respectively."**

*20. Line 246: Confusing. These are individual pixels within the full spatial illumination of a laser? And are these not correlated with the bad pixels mentioned earlier?*

Lines 245-250 of the original manuscript have been revised to clarify the observed anomalous behavior of the ISRFs.

**"After applying the Savitzky-Golay filter to the ISRF across all spatial and spectral pixels, a small number of ISRFs (74 out of 7767 in the CH$_4$ band, 87 out of 8630 in the O$_2$ band) exhibit anomalous widths by way of sharp contrasts with their neighbors, presumably due to insufficient bad pixel removal. By nature, the ISRF shape should vary smoothly between spatial and spectral pixels. Due to the sparse ISRF measurements in the spectral dimension, it is desirable to remove those outliers to avoid inference to a broad wavelength range."**

*21. Section 7 : Please give dates and lengths of flight. Date of flight can only be read from caption of Fig. 17.*

The following sentence specifying the date and time of flight has been inserted after the opening sentence of Section 7 (line 295 of the original manuscript).

**"The flight coverage began at 15:51:29 UTC and ended at 19:40:46 UTC on 11/8/2019."**

*22. Line 303: Why was only a single across track position done? This could be repeated for similar across track positions at other points during the flight. Are results similar?*

The spectral fitting with varying ISRF width scaling factor has been done for all across-track positions in this research flight. Variations in both across- and along-track dimensions were observed, and the updated ISRF is used in the final level 2 retrieval. However, discussion of those results would require too much information on the retrieval algorithm implementation and is deemed to be out of scope of this paper. We are preparing manuscripts dedicated to the in-flight performance and retrieval algorithm. The sentence at line 317 is changed to:

**"The spectral fitting with varying ISRF width is applied to other across-track positions throughout the flight and reveals across-track and time dependent ISRF changes."**

*23. Line 308 : Difference is attributed to temperature changes due to environment. Could the temperature difference be quantified?*

The internal temperature of the instrument was not monitored in flight. We did observe correlation between cabin temperature and ISRF changes. The sentence in lines 309-310 is modified to

**"The small F-number of both spectrometers makes the focal point sensitive to deformation of the mechanical structure. This may arise from a difference from the lab and flight environments, such as a change in the temperature of the optical bench or mechanical stress of the instrument that responds to cabin temperature and/or aircraft motion."**

---

## Author Comment (AC2)

Response to Referee #2:

We appreciate the very helpful feedback from the referee. The referee's comments are listed in *italics*, followed by our response in blue. New/modified text in the manuscript is in **bold**.

*1. Line 33: Do the authors mean to say 200 km swath rather than what appears to be an area? Please provide the single-dimension width.*

Upon re-checking, the swath at nadir is 260 km (not km$^2$). The sentence at lines 32-33 in the original manuscript has been updated to reflect this:

**"MethaneSAT aims to characterize oil and gas basin-scale, diffuse CH$_4$ emissions through a wide swath of 260 km and at the same time locate and quantify large point sources within each target area that is typically 200 × 140 km$^2$ (Benmergui et al., 2020; MethaneSAT, LLC, 2020).**

*2. Line 115: Assuming that the camera images the sphere aperture (i.e. the sphere is not being used as an irradiance source), how can changing the diameter of the output port alter the sphere radiance (it will alter its irradiance)?*

The sentence at line 115 has been revised to clarify.

**"During the MethaneAIR flat field measurements, the light level was tuned from zero to just beyond detector saturation in 40 steps by adjusting the variable input aperture between the integrating sphere and the lamp."**

*3. Line 117: The description of 'calibration curves' lacks clarity. If you're going to talk about fits you need to describe exactly what you're fitting.*

The following text has been added to line 117 for clarification.

**"This resulted in curves of spectral radiance (phot s$^{-1}$ cm$^2$ sr$^{-1}$ nm$^{-1}$) vs. dark-subtracted focal plane intensity (DN s$^{-1}$) for each exposure time and every active pixel. These radiometric calibration curves were fitted by fifth-order polynomials with the intercept forced to be zero."**

*4. Line 186: Please be more specific what you mean by 'percent of total detected light.' Is this the modeled SL for a simulated input scene? Is the percentage measured relative to the useful signal at each wavelength? What does 'total detected light' mean?*

This is the percentage of stray light at the far field relative to the useful signal at each spatial/spectral position and is calculated by summing up the far-field kernel. To clarify, the sentence at line 186 has been removed, and the following sentence is added to line 177:

**"The sum of the far-field kernel is 2.4% for the CH$_4$ band and 2.1% for the O$_2$ band. This indicates that the stray light is small relative to the useful signals at each spatial and spectral position."**

*5. Line 262: No mention is made of a radiometric correction to the ISRF. The iterative oversampling described in Section 5.1 yields empirical scale factors to 'stitch' the multiple pixels together. The scale factors remove differences in radiometric response between the adjacent spectral pixels. These differences include PRNU, which you want to remove, but also spectral response variations (because the pixels are at different wavelengths), which you do not want to remove. This latter variation needs to be included in the ISRFs before they can be used in any forward model calculation. I would expect any description of the 3D lookup table to mention how this variation is reintroduced. If the authors have concluded the effect is negligible then they should state so in the text.*

The ISRF variation in this short wavelength range is far smaller than the measurement noise. The overall spectral variation is significant but smooth across the ~100 nm range, so the variation within ±0.1 nm or about 1 spectral sampling interval is negligible. The sentence spanning lines 196-197 has been replaced with the following for clarification:

**"The ISRF variation between spectral pixels within a small wavelength window ($\pm$ 0.1 nm) is negligible."**

---

## Author Comment (AC3)

Response to Referee #3:

We appreciate the very helpful feedback from the referee. The referee's comments are listed in *italics*, followed by our response in blue. New/modified text in the manuscript is in **bold**.

*1. Spectral pixels vs. central wavelength: From line 11 the term central wavelength is taken to be that of the test laser source, where spectral pixel is a coordinate on the 2-D detector (spatial pixel being the other coordinate). However, Fig. 14 seems to use both central wavelength and spectral pixel inter-changeably. The abscissa of the plots is labeled Central wavelength, but the caption mentions spectral pixels..*

"Spectral pixel" does refer to a coordinate on the detector. In the original figure, "central wavelength" was used to label the spectral dimension because it is more meaningful than the spectral pixel index. Fig. 14 has been updated with Gouraud shading, effectively bridging the gaps between the central wavelengths. As such, the abscissa label has been updated to simply "wavelength". The following sentence has been added to the caption of Fig. 14 for clarification.

**"Gouraud shading is applied to render smooth ISRF variation across the FPA. Wavelength labels have been projected to the abscissa in order to provide more context compared to the spectral pixel index."**

*2. Median filtering, Lines 245-256: It is not clear to what dimension of the table in line 250 the median filter is applied.*

The median filter is applied to each element in the 3D table. The input to the filter at each element consists of 5 elements in the spatial dimension, 3 elements in the spectral dimension, and 1 element in the relative wavelength dimension. The sentence at original line 249-250 has been revised, and further clarification has been added.

**"To remove the effects of these remaining anomalous pixels, a median filter was first applied to the spatial and spectral dimensions of all ISRFs, which are assembled to a table defined in spatial, spectral, and relative wavelength dimensions. The median filter window sizes are 5 elements in the spatial and 3 elements in the spectral dimension."**

*3. Line 264: Again confused by the reference to "all spectral pixels" when I thought the ISRF was only measured/derived for specific "central wavelengths" where the laser source operated.*

The ISRF was indeed only derived for specific central wavelengths. The phrase "all spectral pixels" referenced all possible positions in the spectral dimension corresponding to the full wavelength range. The text at original line 264 has been revised for clarification.

**"The spectral variation of the ISRF at ~10 central wavelength positions is smooth, making it possible to interpolate the ISRF along the spectral dimension to all possible wavelengths. However, the spatial variation of the ISRF is significant due to the slit width irregularity."**

*4. Table 2: The footnote #6 is missing, but referenced in the last column.*

The footnotes have been corrected, where "7" is changed to "6".

*5. Line 343: From what was presented in the article it is not definitive that the possible inhomogeneous illumination of the slit produced asymmetric spectral response functions.*

This sentence has been removed, and the sentence following it has been revised:

**"In future stray light measurements, the pinhole will be replaced with a thin slit in order to fully illuminate the width of the spectrometer slit and hence avoid distorting the spectral response."**

---

## Author Comment (AC4)

Response to Referee #4:

We appreciate the very helpful feedback from the referee. The referee's comments are listed in *italics*, followed by our response in blue. New/modified text in the manuscript is in **bold**.

*1. Line 115: How was saturation defined / measured? How uniform is the saturation level across the spatial and spectral dimensions?*

The following text has been added at line 115 in the original manuscript to provide this information.

**"Saturated values were identified by plotting signal level as a function of exposure time and finding the "knee" where the response became nonlinear. For almost all pixels, this occurred within a few hundred counts of 10,000 DN."**

*2. Line 147: Why did the laser power need to be increased at long wavelengths if a gain correction was already applied?*

The ISRF needs to be measured at a higher radiance level to partially compensate for the reduced signal-to-noise ratio due to QE drop. The text has been edited at original line 147:

**"In the $CH_4$ band, the laser power was increased progressively from -3.0 dBm at wavelengths ≤ 1640 nm, up to +2.5 dBm at 1670 nm, in order to maintain high SNR as the QE decreased at longer wavelengths."**

*3. Fig 13: Some ISRFs are not monotonic with relative wavelength. Is this concerning and/or does it have a physical explanation? Also, are asymmetries understood?*

The referenced ISRF features are at approximately the $10^{-3}$ level. Some of those features are inevitable residual noise that cannot be fully removed by the smoothing applied in Fig. 11. We have tried to fit analytical ISRF functions similar to the TROPOMI ISRF to mitigate those non-monotonic features, but could not find an analytical function that can adequately model all spectral and spatial positions. We do not have a physical MethaneAIR ISRF model to understand the asymmetries, which is not that significant. The following is added after the first sentence at line 257:

**"Non-smooth features at $10^{-3}$ level remain over some ISRF tails at log scale due to detector noise that cannot be fully suppressed."**

*4. Fig 17: How small do residuals need to meet science objectives?*

The following sentence about fitting residuals is added to line 317 of the original manuscript:

**"Those fitting residuals are consistent with the signal-to-noise ratio predicted by the MethaneAIR specs. The retrieved XCH$_4$ will be presented in the following algorithm paper."**

*5. Line 318: In addition to retrieving changes, can any in-flight measurements be made to update ISRF in flight?*

MethaneAIR does not have this capability. Further information has been added to line 319 of the original manuscript.

**"Ideally, in-flight measurements from on-board lasers would be used to update the ISRF in flight, as is done for TROPOMI (van Kempen et al., 2019). MethaneAIR and MethaneSAT are not equipped with this capability, but on-orbit ISRF monitoring is being planned by looking at targets on the earth, the airglow, and the moon for MethaneSAT.**